# SARDet-100K: Towards Open-Source Benchmark and ToolKit for Large-Scale SAR Object Detection

**Yuxuan Li**[1]   **Xiang Li**[1,2,†]   **Weijie Li**[3]   **Qibin Hou**[1,2]   **Li Liu**[3]
**Ming-Ming Cheng**[1,2]   **Jian Yang**[1,†]

[1] PCA Lab, VCIP, CS, Nankai University      [2]NKIARI, Futian, Shenzhen
[3]Academy of Advanced Technology Research of Hunan
[†] Corresponding Authors
`yuxuan.li.17@ucl.ac.uk, {xiang.li.implus,houqb,cmm,csjyang}@nankai.edu.cn`

## Abstract

Synthetic Aperture Radar (SAR) object detection has gained significant attention recently due to its irreplaceable all-weather imaging capabilities. However, this research field suffers from both limited public datasets (mostly comprising <2K images with only mono-category objects) and inaccessible source code. To tackle these challenges, we establish a new benchmark dataset and an open-source method for large-scale SAR object detection. Our dataset, SARDet-100K, is a result of intense surveying, collecting, and standardizing 10 existing SAR detection datasets, providing a large-scale and diverse dataset for research purposes. To the best of our knowledge, SARDet-100K is the first COCO-level large-scale multi-class SAR object detection dataset ever created. With this high-quality dataset, we conducted comprehensive experiments and uncovered a crucial challenge in SAR object detection: the substantial disparities between the pretraining on RGB datasets and finetuning on SAR datasets in terms of both data domain and model structure. To bridge these gaps, we propose a novel Multi-Stage with Filter Augmentation (MSFA) pretraining framework that tackles the problems from the perspective of data input, domain transition, and model migration. The proposed MSFA method significantly enhances the performance of SAR object detection models while demonstrating exellent generalizability and flexibility across diverse models. This work aims to pave the way for further advancements in SAR object detection. The dataset and code is available at `https://github.com/zcablii/SARDet_100K`.

## 1   Introduction

Synthetic Aperture Radar (SAR) [57; 60]is a pivotal technology in remote sensing, providing numerous advantages over traditional optical sensors. Notably, SAR possesses the capability to acquire geographical images under any weather conditions, irrespective of factors such as sunlight, land cover, or certain types of camouflage, as demonstrated in Fig. 1(a). As a consequence of these advantages, SAR has found extensive applications in critical domains, including national defence [48], humanitarian relief [3; 68], camouflage detection [19], and geological exploration [51; 24].

With its invaluable benefits, the field of SAR object detection has garnered increasing attention. In recent years, there has been a substantial increase in the number of research papers focusing on this field, as illustrated in Fig. 1(b). Despite the increasing influence, this research area has suffered from significant challenges including limited resources and transferring gaps.

**Limited resources.**   A significant obstacle in high-resolution SAR image object detection is the sensitivity of SAR images, coupled with the high costs associated with annotating these images. This

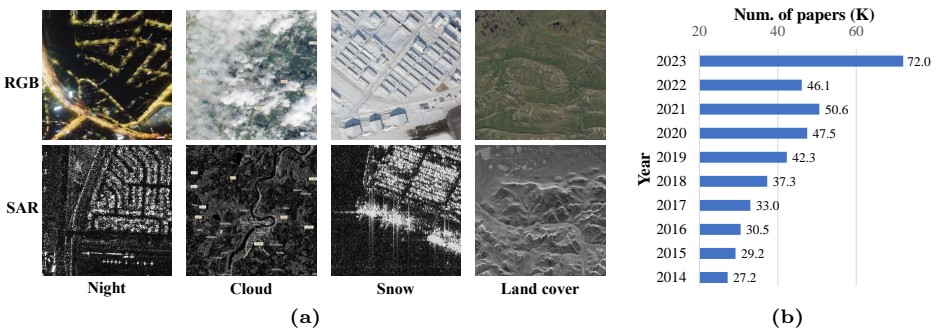

Figure 1: **(a)** Advantages of SAR image: independent of weather conditions, sunlight and land cover. **(b)** Number of papers (thousands) retrieved from Google Scholar using keywords "SAR Detection".

severely restricts the availability of public datasets. Existing datasets, such as SAR-AIRcraft [90], Air-SARShip [76], SSDD [84], and HRSID [71], typically consist of a singular type of object against a simplistic background. Moreover, these datasets are generally limited in scale, potentially introducing bias when evaluating different methodologies. Additionally, a notable barrier to advancing research in SAR object detection is the lack of publicly accessible source code, making it challenging to reproduce previous research findings and conduct fair comparisons or build upon existing work.

To address this problem, we merge the most publicly available SAR detection datasets. This effort includes a comprehensive review of current public SAR detection resources, followed by the collection and standardization of these datasets into a uniform format, creating a unified large-scale multi-class dataset for SAR object detection, named SARDet-100k. This dataset comprises approximately 117k images and 246k instances of objects across six distinct categories. To our knowledge, SARDet-100k is the first dataset of COCO-scale magnitude in this research area. It significantly contributes to overcoming the previously mentioned limitations by providing a rich resource for the development and evaluation of SAR object detection models. Moreover, the dataset and source code will be made publicly available.

**Transferring gaps.** Through our empirical research and detailed analysis, we have identified that a principal hurdle in SAR object detection is the significant domain gap and model gap encountered when transferring a backbone network pretrained on natural RGB datasets (e.g., ImageNet [17]), to a detection network on SAR imagery. The domain gap stems from the stark visual discrepancies between RGB and SAR imagery, whereas the model gap arises from the model differences between the pretrained backbone and the whole detection framework employed in the downstream task.

To mitigate the aforementioned domain gap and model gap, we propose a novel Multi-Stage with Filter Augmentation (MSFA) pretraining framework to bridge these gaps. This framework addresses the challenge from multiple angles: data input, domain transition, and model migration, each tailored to the unique properties of the SAR image detection task. For data input: to address the input domain gap between the pretrain and finetune datasets, we employ traditional, handcrafted feature descriptors. These descriptors efficiently transform the input data from pixel space to a feature space that is not only robust to noise but also statistically narrows the gap between data from RGB and SAR modalities (see Fig. 2(a)), thereby enhancing the transferability of pretrained knowledge. For domain transition: we propose a domain transition bridge utilizing an optical remote sensing detection dataset. This bridge connects natural RGB images through optics correlation and SAR images through object correlation, establishing a hierarchical pretraining approach that effectively closes the domain gap between RGB and SAR imagery (see Fig. 2(b)). For model migration: to guarantee thorough training of the entire detection framework and to facilitate complete model migration for finetuning, we employ the entire detector as a bridging model throughout the multi-stage pretraining process.

The MSFA framework demonstrates remarkable efficacy in reducing the substantial domain and model gaps typically encountered between the pretraining and finetuning stages. MSFA is not only effective but also general and applicable across various modern deep neural networks.

Our contribution to the field of SAR object detection can be concluded into the following FOUR points:

- Introduction of the first COCO-level large-scale dataset for SAR multi-category object detection.
- Identification of critical gaps in traditional model pretrain and finetune approaches for SAR object detection.
- Proposal of a Multi-Stage with Filter Augmentation (MSFA) pretraining framework, which demonstrates remarkable effectiveness, as well as excellent generalizability and flexibility across various deep network models.
- Establishment of a new benchmark in SAR object detection by releasing the datasets and code associated with our research. This contribution is expected to foster further advancements and progress in the field.

## 2 Related Work

### 2.1 SAR Images and Handcraft Features

SAR imaging often suffers from poor image quality due to multiplicative speckle noise and artifacts [57; 47]. To mitigate this, many traditional handcrafted feature descriptors have been developed or adapted to extract more discernible features from SAR images. These include Histogram of Oriented Gradients [15] (HOG), Canny Edge Detector [5], Gradient by Ratio Edge (GRE) [29], Haar-like [62] Feature Descriptor and Wavelet Scattering Transform [45] (WST). Early works employed traditional algorithms, such as HOG for SAR object recognition [56; 49], Canny [28; 38] for edge detection. However, in recent years, the field of SAR image analysis has been largely dominated by deep learning approaches.

While recent studies have focused on tasks related to low-level processing [69; 86], classification [83; 85; 93; 26; 50; 78] and pretrain [29; 29], they have attempted to integrate classic handcrafted features into modern neural networks for robust SAR image feature extraction and refinement. In contrast, our work does not simply inject such handcrafted features into networks, but explores the benefits and potentials of handcrafted features in domain adaptation and SAR object detection under modern deep neural networks. This research area remains largely unexplored, and our work aims to bridge this gap.

### 2.2 SAR Object Detection

Various popular deep learning-based object detection frameworks, including RetinaNet [33], FCOS [59], GFL [30], RCNN series [53; 4], YOLO series [52; 10], and DETR [6], demonstrate remarkable generalizability in the field of general object detection. Additionally, modern backbone networks such as ConvNext [42], VAN [22], LSKNet [31] and Swin Transformer [41] are designed to efficiently and effectively model visual features. However, SAR image object detection poses unique challenges due to factors such as small object size, speckle noise, and sparse information inherent in SAR images. As a result, recent deep learning methods for SAR object detection primarily focus on network and module design to address these challenges. Approaches like MGCAN [9], MSSDNet [91], and SEFEPNet [79] enhance object features through multiscale feature fusion. Quad-FPN [82] combines four distinct feature pyramid networks for thorough multiscale feature interaction to alleviate noise interferences and multi-scale object feature misalignment. PADN [88] and EWFAN [65] employ attention mechanisms to enhance object features in the presence of SAR speckle noise. CenterNet++ [21], an extension of CenterNet [92], incorporates feature enhancement, multi-scale fusion, and head refinement modules to improve the detector's robustness specifically for SAR images. Additionally, CRTransSar [74], built on the high-performance Swin transformer [41], leverages context representation learning to enhance object features.

While most existing works concentrate on mitigating SAR speckle noise interference through network structure improvements, few attempts to address the issue at the level of input data. Furthermore, most studies utilize ImageNet pretrained backbones as the initialization of the detection framework, overlooking the substantial domain gap between the pretrained nature scenes dataset and the finetuned SAR dataset, as well as the model gap between the backbone and the entire detection framework. Instead, we seek to address these unique challenges through a carefully designed pretraining strategy.

Table 1: Image and instance level statistics of SARDet-100K dataset. *: Origin datasets are cropped into 512 × 512 patches. Ins: Instances, Img: Images.

| Dataset | Images | | | | Instances | | | | Ins/Img |
|---|---|---|---|---|---|---|---|---|---|
| | Train | Val | Test | ALL | Train | Val | Test | ALL | |
| AIR_SARShip 1* [76] | 438 | 23 | 40 | 501 | 816 | 33 | 209 | 1,058 | 2.11 |
| AIR_SARShip 2 [76] | 270 | 15 | 15 | 300 | 1,819 | 127 | 94 | 2,040 | 6.80 |
| HRSID [71] | 3,642 | 981 | 981 | 5,604 | 11,047 | 2,975 | 2,947 | 16,969 | 3.03 |
| MSAR* [75] | 27,159 | 1,479 | 1,520 | 30,158 | 58,988 | 3,091 | 3,123 | 65,202 | 2.16 |
| SADD [80] | 795 | 44 | 44 | 883 | 6,891 | 448 | 496 | 7,835 | 8.87 |
| SAR-AIRcraft* [90] | 13,976 | 1,923 | 2,989 | 18,888 | 27,848 | 4,631 | 5,996 | 38,475 | 2.04 |
| ShipDataset [67] | 31,784 | 3,973 | 3,972 | 39,729 | 40,761 | 5,080 | 5,044 | 50,885 | 1.28 |
| SSDD [84] | 928 | 116 | 116 | 1,160 | 2,041 | 252 | 294 | 2,587 | 2.23 |
| OGSOD [63] | 14,664 | 1,834 | 1,833 | 18,331 | 38,975 | 4,844 | 4,770 | 48,589 | 2.65 |
| SIVED [35] | 837 | 104 | 103 | 1,044 | 9,561 | 1,222 | 1,230 | 12,013 | 11.51 |
| **SARDet-100k** | 94,493 | 10,492 | 11,613 | 116,598 | 198,747 | 22,703 | 24,023 | 245,653 | 2.11 |

Table 2: SARDet-100K source datasets information. GF-3: Gaofen-3, S-1: Sentinel-1. Target categories S: ship, A: aircraft, C: car, B: bridge, H: harbour, T: tank.

| Datasets | Target | Res. (m) | Band | Polarization | Satellites | License |
|---|---|---|---|---|---|---|
| AIR_SARShip [76] | S | 1,3m | C | VV | GF-3 | - |
| HRSID [71] | S | 0.5∼3m | C/X | HH, HV, VH, VV | S-1B,TerraSAR-X,TanDEMX | GNU General Public |
| MSAR [75] | A, T, B, S | ≤ 1m | C | HH, HV, VH, VV | HISEA-1 | CC BY-NC 4.0 |
| SADD [80] | A | 0.5∼3m | X | HH | TerraSAR-X | - |
| SAR-AIRcraft [90] | A | 1m | C | Uni-polar | GF-3 | CC BY-NC 4.0 |
| ShipDataset [67] | S | 3∼25m | C | HH, VV, VH, HV | S-1,GF-3 | - |
| SSDD [84] | S | 1∼15m | C/X | HH, VV, VH, HV | S-1,RadarSat-2,TerraSAR-X | Apache2.0 |
| OGSOD [63] | B, H, T | 3m | C | VV/VH | GF-3 | - |
| SIVED [35] | C | 0.1,0.3m | Ka,Ku,X | VV/HH | Airborne SAR synthetic slice | - |

# 3 A New Benchmark Dataset for SAR Object Detection

## 3.1 Current Status

SAR images are typically captured by satellites, and there is a wealth of low-resolution SAR imagery available, often with a Ground Sample Distance (GSD) of 10m × 10m or larger. Platforms like Sentinel-1 [12] provide access to these images, which offer a macroscopic view of various geophysical places such as cities, mountains, rivers, and cultivated land. This makes them particularly advantageous for scene classification tasks. However, the inherent low resolution of these images constrains their capability to delineate fine details of smaller objects, such as ships, cars, and airplanes. Conversely, high-resolution SAR images provide more detailed information but require significant hardware resources. Moreover, these high-definition images often encompass sensitive information, making them unsuitable for public release. Furthermore, acquiring high-resolution SAR datasets can be very expensive, posing significant challenges to their accessibility.

Numerous research teams frequently encounter budgetary limitations that restrict their capacity to obtain a large and diverse collection of high-resolution SAR datasets. These financial constraints not only limit the scope of geographical areas that can be covered but also affect the variety of data sources that can be accessed. Consequently, the datasets made available by these groups often lack diversity, particularly in aspects such as spectral bands, polarization, and resolution. From a researcher's perspective, evaluating models on such small and homogeneous datasets can introduce bias and lead to unfair performance comparisons.

## 3.2 SARDet-100K

To address the aforementioned challenges, we undertake a thorough survey of SAR object detection datasets. As a result, we carefully collect a total of **10** publicly available high-quality datasets that are not only diverse but also have no conflicting object categories. These data are released by or collected from different countries and institutions, such as scientific research departments in China, space departments in Europe, and military departments in the United States. Detailed information

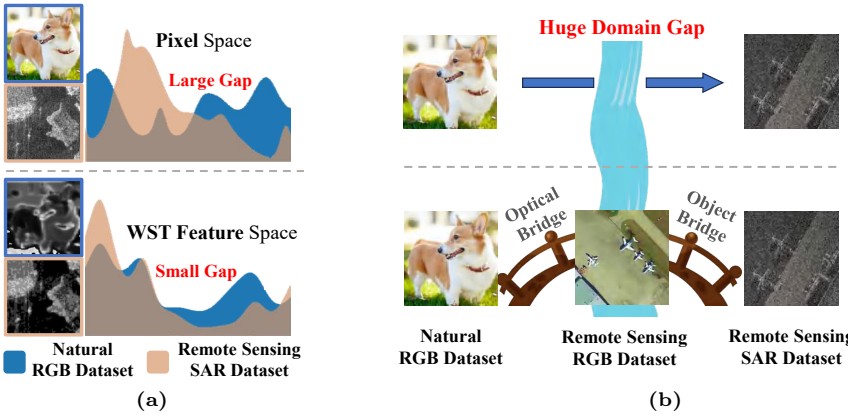

Figure 2: Illustration of the significant domain gap exists between Nature RGB dataset and remote sensing SAR datasets. **(a)** showcases the WST feature space significantly narrows the domain gap. **(b)** demonstrates that the remote sensing RGB dataset serves as an effective domain transit bridge, facilitating smoother domain transfer.

on the collected datasets is shown in Table. 2. To ensure consistency across the collected datasets, we invest considerable time and effort in rigorous dataset standardization. This involves addressing variations in train-val-test splitting status, image resolutions, and annotation formats. More details about data collection and standardization can be found in the Appendix.

Table 1 presents the standardized sub-datasets of SARDet-100K along with their corresponding statistics, which includes information on both image-level and instance-level statistics. The SARDet-100K dataset encompasses a total of 116,598 images, and 245,653 instances distributed across six categories: Aircraft, Ship, Car, Bridge, Tank, and Harbor. SARDet-100K dataset stands as the first large-scale SAR object detection dataset, comparable in size to the widely used COCO [34] dataset (118K images), which is the standard benchmark for general object detection. The scale and diversity of the SARDet-100K dataset effectively simulate real-world scenarios encountered in the application of SAR object detection models across multiple data sources. SARDet-100K provides researchers with robust training and evaluation for advancing SAR object detection algorithms and techniques, fostering the development of SOTA models in this domain.

# 4    Multi-Stage with Filter Augmentation Pretraining Framework

Several recent studies [26; 78; 74; 21] have demonstrated the effectiveness of mature handcrafted features and specialized network module designs in improving SAR object detection performance. However, most of these works rely on the default ImageNet pretraining approach, thus overlooking the significant domain gap between the pretrained nature scenes dataset and the finetuned SAR dataset. Additionally, they fail to address the model gap that exists between the backbone and the entire detection framework. To address these limitations, we propose a novel framework called the Multi-Stage with Filter Augmentation (MSFA) Pretraining Framework. Our framework tackles the challenges from the perspective of data input, domain transition, and model migration. MSFA comprises two core designs: the Filter Augmented Input and the Multi-Stage pretrain strategy.

## 4.1    Filter Augmented Input

As discussed in the *Related Work* section, numerous existing handcrafted feature descriptors leverage meticulously designed filters to extract features. These features, robust and rich in information, act as augmented information derived from the original image. Thus, we propose employing such features as auxiliary information alongside the original pixel data. The filter augmented feature $M$ of data $x$ can be generally defined as:

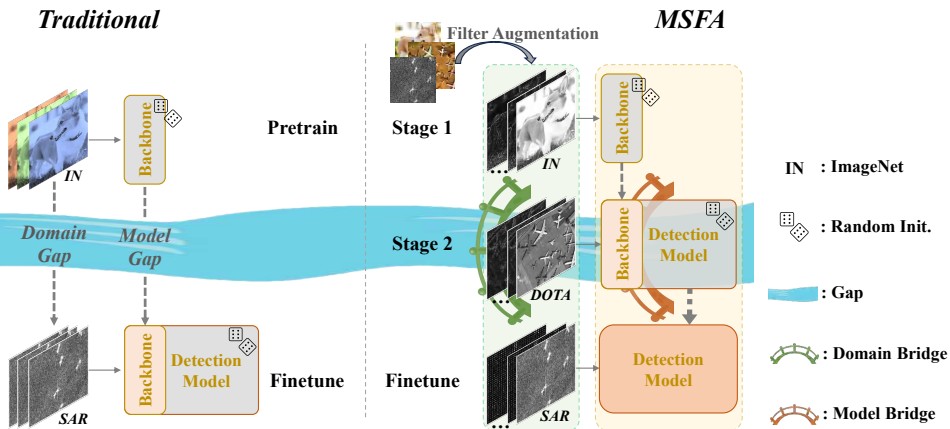

Figure 3: Conceptual illustration of traditional ImageNet pretrain and our proposed Multi-Stage with Filter Augmentation (MSFA) pretrain framework.

$$M_i^x = T_i(x), i \in \{HOG, Canny, Haar, WST, GRE\}. \tag{1}$$

Where $T_i$ is a pre-defined transformation. Drawing inspiration from the information residual design in ResNet [23], we construct the Filter Augmented Input to the detection model, $Inp$, by concatenating the original grayscale SAR image $x$ with the generated filter augmented feature $M_i^x$ as:

$$Inp = \text{concat}(x, M_i^x). \tag{2}$$

By casting the original data input from the heterogeneous pixel space to a homogeneous filter augmented feature space, the domain gaps between different image domains can be greatly reduced, as illustrated in Fig. 2(a).

### 4.1.1 Multi-stage pretrain

We formulate the traditional pretrain schema as:

$$B = \text{Train}_{cls}(B_\theta)(D_{IN}), \tag{3}$$

$$A = \text{Train}_{det}(A_B)(D_{SAR}). \tag{4}$$

The function $\text{Train}_t(a)(b)$ means training a model $a$ on a dataset $b$ with a task $t$, and it returns a trained model. $t$ is the training task, $t \in \{cls, det\}$ where $cls$ stands for classification and $det$ for detection. $B$ indicates the backbone model and $A$ is the whole detection model. Traditionally, the pretrain stage will randomly initialize the backbone model $B_\theta$ and train on the dataset ImageNet $D_{IN}$ (as Eq. (3)). Then finetune on the SAR dataset $D_{SAR}$ with pretrained backbone initialised from detection model $A_B$ (as Eq. (4)).

Our proposed multi-stage pretrain strategy, alternatively, can be illustrated as in the Eq. (3) (5) (6).

$$A' = \text{Train}_{det}(A_B)(D_{RS}). \tag{5}$$

$$A = \text{Train}_{det}(A_{A'})(D_{SAR}). \tag{6}$$

Where an extra second stage pretraining in Eq. (5) is added. We propose the utilization of a large-scale optical remote sensing dataset, $D_{RS}$, as a detection pretrain for domain transit. The dataset consists of optical modal imagery, which also shares similar object shapes, scales, and categories in downstream SAR datasets. This characteristic serves as a valuable bridge between the optical distribution of natural images in ImageNet and the object distribution in SAR remote sensing images. By leveraging such second stage pretrain, the domain gap is effectively minimized, as illustrated in Fig. 2(b).

#### 4.1.2  MSFA

Finally, the proposed MSFA framework integrates Filter Augmented Input with Multi-Stage pretraining, as illustrated in Fig. 3. Our MSFA framework effectively bridges the substantial domain and modal gaps between pretraining on nature images and finetuning on SAR image detection.

By introducing Filter Augmented Input, we leverage mature handcrafted feature descriptors to extract noise-robust features. This also enables us to effectively transform the heterogeneous image domains of both pretraining and finetuning images into a homogeneous feature domain. By unifying the input data into a consistent feature domain, we address the disparities that exist between different types of images. Consequently, it enhances the alignment and transferability of knowledge across domains. Moreover, the incorporation of multi-stage training involves utilizing an additional large-scale optical remote sensing dataset for detection pretraining. This dataset acts as a domain bridge, connecting the domain of ImageNet's nature images with that of SAR remote sensing images. As a result, it further reduces the domain gaps, facilitating a smoother transition between the two domains. Furthermore, the detection pretraining in the second stage of the MSFA framework can also act as a model bridge. It allows for the comprehensive training of the entire detection framework, rather than solely focusing on the backbone, making the whole detection framework well-initialized to perform optimally in the SAR detection finetuning.

## 5  Experiments and Analysis

### 5.1  Filter Augmented Input

To investigate and assess the impact of the proposed Filter Augmented Input, we conduct experiments on each traditional feature descriptor discussed in the *Related Work*, within the

| Input | mAP ↑ | mAP$_{50}$ ↑ |
|---|---|---|
| SAR (as RGB) | 50.2 | 83.0 |
| SAR+Canny | 50.7 | 83.6 |
| SAR+Hog | 50.7 | 83.5 |
| SAR+Haar | 50.6 | 83.4 |
| SAR+WST | **51.1** | 83.9 |
| SAR+GRE | 50.6 | 83.8 |
| SAR+Hog+Haar+WST | **51.1** | **84.0** |

Table 3:  Comparison of different Filter Augmented Inputs using Faster R-CNN and ResNet50 as the detection model.

| Domain | PCC ↑ |
|---|---|
| Pixel Space | 0.394 |
| Canny Space | 0.992 |
| Hog Space | 0.995 |
| Haar Space | 0.990 |
| WST Space | **0.996** |
| GRE Space | 0.984 |

Table 4:  Pearson Correlation Coefficients (PCC) of ImageNet and SARDet-100k on RGB and handcrafted feature spaces.

framework of our proposed MSFA method. The findings, detailed in Table 3, indicate that incorporating these handcrafted features notably enhances the performance of the detector. Additionally, our analysis reveals that converting image pixels into handcrafted feature spaces significantly minimizes the distributional gaps between the ImageNet and the SARDet-100K datasets. This is particularly evident in the Pearson Correlation Coefficients (PCC) between the inputs of the ImageNet and SARDet-100K datasets, as illustrated in Table 4. This underscores the efficacy of the proposed approach in bridging the domain gap between natural and SAR images, thereby enhancing the efficiency of knowledge transfer from the pretraining process.

Remarkably, the Wavelet Scattering Transform (WST) feature stands out for its exceptional performance. This superiority can be attributed not just to its role in significantly narrowing the domain gap, but also to its capacity for extracting rich, multi-scale information. Such information acts as a robust auxiliary feature by mitigating noise and preserving object-related details. However, we also find that using multiple filter augmented features will not have further significant performance gain. It is possible that the existing WST already captures the essential information necessary for effective object detection, and incorporating additional ones does not provide substantial additional beneficial information.

Due to the outstanding performance of WST, we employed it as the default Filter Augmented Input in our MSFA method for the remainder of the paper.

Table 5: Comparison of different pretrain strategies using Faster-RCNN and ResNet50 as the detection model.

| ID | Model Input | Pretrain | | | mAP ↑ |
|----|-------------|----------|---------|-----------|-------|
| | | Multi-stage | Dataset | Component | |
| 1 | SAR (Raw Pixels) | ✗ | ImageNet | Backbone | 49.0 |
| 2 | | ✓ | ImageNet + DIOR | Framework | 49.5 |
| 3 | | ✓ | ImageNet + DOTA | Backbone | 49.3 |
| 4 | | | | Framework | **50.2** |
| 5 | SAR+WST (Filter Augmented) | ✗ | ImageNet | Backbone | 49.2 |
| 6 | | ✓ | ImageNet + DIOR | Framework | 50.1 |
| 7 | | ✓ | ImageNet + DOTA | Backbone | 49.6 |
| 8 | | | | Framework | **51.1** |

## 5.2 Multi-stage Pretrain

To evaluate the effectiveness of the proposed multi-stage pretraining approach, we conduct experiments in which we keep the input modality consistent and finetuned the detection model using various pretraining strategies on the SARDet-100K dataset. As a baseline, Exp. 1 takes single-channel SAR data as input, pretrains the backbone model on ImageNet for 100 epochs, and then directly finetune the detector on the SARDet-100K dataset (following the widely used default setting). In addition to the baseline, we perform a second stage pretrain specifically for object detection on optical remote sensing datasets, such as DOTA [73] or DIOR [27]. (Details on DOTA and DIOR datasets can be found in the Appendix). Following the second pretraining stage, we finetune the model either solely on the backbone or on the entire framework.

The results of Exp. 2, 4, 6, and 8 in Table 5 prove the substantial advantages of the two-stage pretraining approach. Notably, even the relatively small-scale DIOR dataset showcases noticeable performance gains compared to the baseline (Exp. 1 and 5). This observation underscores the significance of reducing the domain gap during the pretraining phase of SAR detection.

However, the DIOR dataset pretraining is not as effective as the larger-scale DOTA dataset (Exp. 2 vs 4, 6 vs 8). This comparison underscores the significance of the pretraining scale in achieving optimal results. The DOTA dataset, with its larger scale and similar average instance area to SARDet-100K, provides a more comprehensive and informative pretraining, leading to improved performance during the subsequent finetuning stage.

The comparison between Exp. 3 & 4 and Exp. 7 & 8, demonstrate the superiority of pretraining the entire framework over solely the backbone, highlighting the significant impact of the model gap on the performance of SAR detection.

In summary, our proposed multi-stage pretraining strategy in MSFA alleviates both the data domain gap and the model gap between pretraining and the downstream model, leading to significant enhancements in SAR detection performance. Detailed experimental results and visualization are available in the Appendix.

## 5.3 Generalizability of MSFA

To assess the effectiveness and generalizability of the proposed MSFA, we conduct experiments using various detectors and backbones, as presented in Fig. 4(a) and 4(b). Significant performance improvement is observed across different frameworks (including single-stage [53; 4; 43], two-stage [33; 30; 59], and end2end [94; 58; 39]) and diverse backbones (including ResNets [23], ConvNexts [42], VANs [22], and Swin-Transformer [41] networks). It provides strong evidence for the effectiveness and wide applicability of our proposed method. Furthermore, we observe stable performance improvements as we scale up the backbone size, as shown in Fig. 4(b), indicating the good scalability of our proposed method.

Significantly, the design of our MSFA method was developed with flexibility, generalizability and wide applicability in mind. Therefore the method can be seamlessly integrated into most existing models without any modifications.

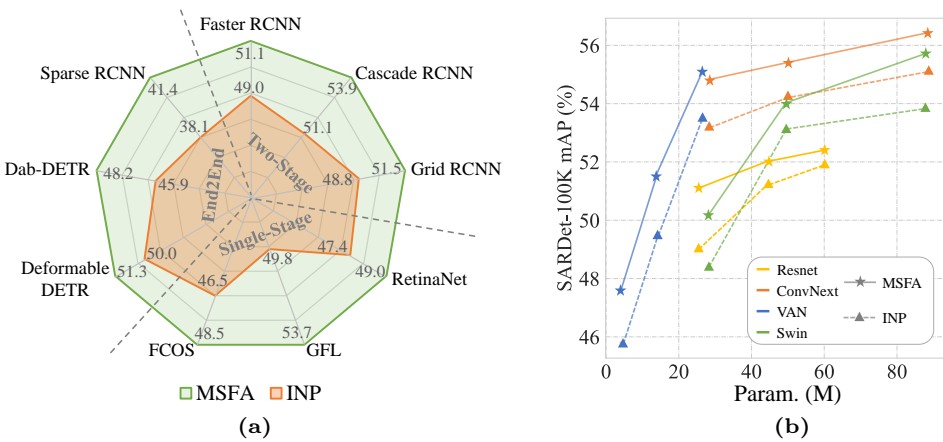

Figure 4: Generalization of MSFA on different detection frameworks **(a)** and different backbones **(b)**. Models are finetuned and tested on SARDet-100K dataset. INP: Traditional ImageNet Pretrain on backbone network only.

Table 6: Comparison of the proposed MSFA with previous state-of-the-art methods on SSDD and HRSID datasets.

| Detectors | | Open Source | Year | mAP$_{50}$ ↑ | |
|---|---|---|---|---|---|
| | | | | SSDD | HRSID |
| General Detectors | Grid R-CNN [43] | ✓ | 2019 | 88.9 | 79.4 |
| | Faster R-CNN [53] | ✓ | 2015 | 89.7 | 80.7 |
| | Cascade R-CNN [4] | ✓ | 2019 | 90.5 | 81.3 |
| | Free-Anchor [87] | ✓ | 2019 | 91.0 | 81.8 |
| | Double-Head R-CNN [72] | ✓ | 2020 | 91.1 | 82.1 |
| | PANET [40] | ✓ | 2018 | 91.2 | 81.6 |
| | DCN [14] | ✓ | 2017 | 92.3 | 82.1 |
| SAR Detectors | NNAM [7] | ✗ | 2019 | 79.8 | - |
| | DCMSNM [25] | ✗ | 2018 | 89.6 | - |
| | ARPN [89] | ✗ | 2020 | 89.9 | 81.8 |
| | DAPN [13] | ✗ | 2019 | 90.6 | 81.8 |
| | HR-SDNet [70] | ✗ | 2020 | 90.8 | 82.5 |
| | SER Faster R-CNN [37] | ✗ | 2018 | 91.5 | 81.5 |
| | FBR-Net [20] | ✗ | 2020 | 94.1 | - |
| | NRENet [44] | ✗ | 2024 | 94.6 | 75.6 |
| | CenterNet++ [21] | ✗ | 2021 | 95.1 | - |
| | CRTransSar [74] | ✗ | 2022 | 97.0 | - |
| | SARATR-X [77] | ✗ | 2024 | 97.3 | 80.3 |
| Faster R-CNN + VAN-B | | ✓ | 2023 | 92.9 | 81.8 |
| **MSFA** (Faster R-CNN + VAN-B) | | ✓ | 2024 | **97.9**(+5.0) | **83.7**(+1.9) |

## 5.4 Comparison with SOTAs

We compare various SOTA methods, including both general object detection models [43; 53; 14; 64; 4; 87; 72], as well as SAR object detection models [7; 25; 89; 13; 70; 40; 37; 74; 21; 20]. We evaluate their performance on the SSDD and HRSID datasets, which are the commonly used benchmarks for SAR object detection. To leverage the superior efficiency and performance of the VAN [22] backbone (as shown in Fig. 4(b)), we employ the classic faster R-CNN detection framework with the lightweight VAN-B (Param. 26.6M) backbone as our detection model. The results presented in Table 6 demonstrate that our MSFA method outperforms all the compared methods by a significant margin. Specifically, MSFA achieves a mAP@50 of 97.9% on the SSDD dataset and a mAP@50 of 83.7% on the HRSID dataset, setting new state-of-the-art results. It is noteworthy that our method is the only open-sourced method among the compared SAR detection SOTAs.

# 6 Limitaion and Future Work

The scope of this paper is limited to supervised pretraining. However, considering the availability of a vast amount of unannotated SAR images, it would be valuable to explore the potential of semi-supervised, weakly-supervised or unsupervised learning methods for domain transfer in SAR object detection.

While this paper aims to propose a simple, practical, effective, and generic method, it does not delve into the details of specific designs. Future work can be expanded to explore the aforementioned directions in more depth, incorporating intricate and specialized designs to enhance the performance and capabilities of SAR object detection.

# 7 Conclusion

This paper presents a new benchmark for large-scale SAR object detection, introducing the SARDet-100k dataset and the Multi-Stage with Filter Augmentation (MSFA) pretrain method. Our SARDet-100k dataset comprises over 116K images spanning 6 categories, providing a large and diverse dataset for conducting SAR object detection research. To bridge the domain and model gaps between pretraining and finetuning stages in SAR object detection, we propose the MSFA pretraining framework. MSFA significantly enhances the performance of SAR object detection models, setting new state-of-the-art performance in previous benchmark datasets. Moreover, MSFA demonstrates remarkable generalizability and flexibility across various models. Our research endeavours to overcome the current obstacles prevalent in SAR object detection. We anticipate our contributions will pave the way for future research and innovations in this domain.

Our research endeavours to overcome the current obstacles prevalent in SAR object detection. We anticipate our contributions will pave the way for future research and innovations in this domain.

# 8 Acknowledgement

**We extend our deepest gratitude to the following researchers, listed alphabetically by first name: Hong Zhang, Runfan Xia, Shunjun Wei, Tianwen Zhang, Xian Sun, Xiaofang Zhu, Xiaoling Zhang, and other contributing researchers, for allowing us to integrate their datasets. Their contributions have greatly advanced and promoted research in this field.**

This work is supported by the National Science Fund of China (62361166670,62276145, 62176130, 62206134) and the Fundamental Research Funds for the Central Universities (Nankai University: 070-63233084, 070-63243142). Computation is supported by the Supercomputing Center of Nankai University (NKSC).

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

# A Appendix

## A.1 SARDet-100K Dataset Visualization

Fig. S5 offers a visualization of sample images from the proposed SARDet-100K dataset. Representative samples for each category, including Ship, Tank, Bridge, Harbour, Aircraft, and Car, are showcased.

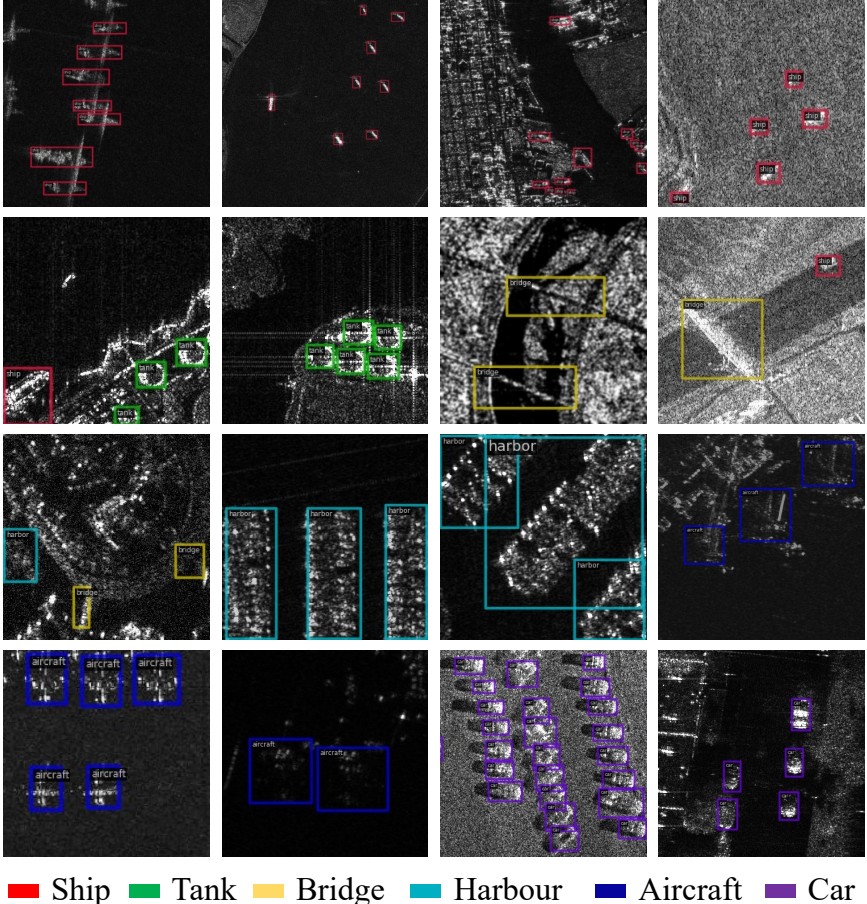

Figure S5: Visualization of sample images from the proposed SARDet-100K dataset.

## A.2 Standardization.

We collect a total of **10** publicly available high-quality datasets that are not only diverse but also have no conflicting object categories. These data are released by or collected from different countries and institutions, such as scientific research departments in China, space departments in Europe, and military departments in the United States. To ensure consistency across the collected datasets, it is necessary to undertake standardization processes. This involves addressing variations in train-val-test splitting status, image resolutions, and annotation formats. The overview of SARDet-100K dataset process pipeline is illustrated in Fig. S6(a).

If the source dataset already provides predefined train, validation, and test splits, we adopt their split settings. Otherwise, we perform the splitting by ensuring a ratio of 8:1:1 for the train, validation, and test sets respectively.

Additionally, we tackle the issue of high image resolution in some of the collected datasets. This concern arises because resizing these images before passing them to the model could result in extremely small targets. We perform image slicing on all datasets that contain images larger than

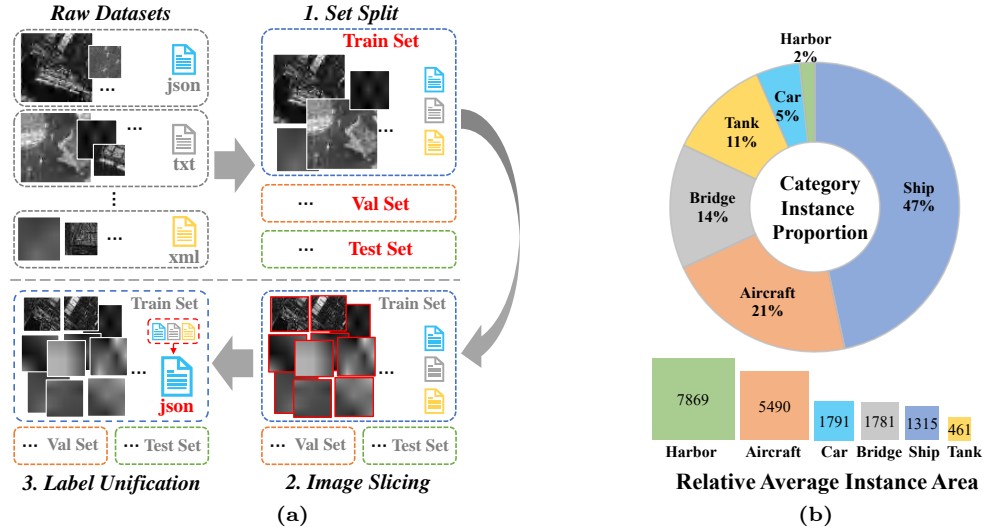

Figure S6: **(a)** SARDet-100K dataset standardization process, encompassing set splitting, large image slicing, and label annotation format unification. **(b)** Percentage of instances for each category and average instance area (in pixels) in SARDet-100K.

$1000 \times 1000$ resolution. Specifically, for AIR SARShip 1, MSAR, and SAR-AIRcraft datasets, we crop each image into patches of size $512 \times 512$ with a patch overlap of 200.

Furthermore, we convert all dataset annotations into the COCO annotation format [34]. This step ensures consistency and compatibility among the different datasets. Consequently, the merged dataset, SARDet-100K, is also standardized in the COCO format, which is readily compatible with popular open-source detection code frameworks, eliminating the need for additional manual data preprocessing. Fig. S6(b) provides an overview of the category-level statistics for the SARDet-100K dataset.

## A.3 Handcrafted Feature Descriptors

**Histogram of Oriented Gradients [15] (HOG).** HOG is a widely used local feature descriptor in image processing and computer vision. It captures and represents the image's local structure and shape information by analyzing the distribution of gradient orientations. HOG is proven to be effective for SAR image classification and object detection tasks because it is invariant to random noise in SAR images. Early works employ HOG for SAR object recognition [56; 49], and recent studies demonstrate the effectiveness of HOG features for SAR image classification [83; 85], as well as the model pretraining [66] in modern neural networks.

**Canny Edge Detector [5].** Canny is a widely used edge detection algorithm, which aims to identify significant edges in an image while minimizing noise and spurious responses. The algorithm utilizes Gaussian smoothing, pixel orientation gradient magnitude, and non-maximum suppression. Early research [28; 38] recognize the advantage of Canny for SAR image processing. Recent works also verify the effectiveness of the Canny feature in SAR image edge detection [69], interference detection [18], and image registration [86]. Other than Canny Edge Detector, Gradient by Ratio Edge (GRE) [29], a recently proposed edge detector, leverages SAR-HOG [16] and SAR-SIFT [56] to achieve effective edge detection in SAR images.

**Haar-like [62] Feature Descriptor.** Haar-like features are commonly used for face detection [46; 2], pedestrian detection [81], and other target detection tasks [32; 36]. They describe the characteristics of an image by utilizing predefined feature templates. Haar-like features can capture linear features, edge features, point features, and diagonal features. Early studies [54; 55] demonstrate the robustness of Haar-like features for SAR object detection. A recent approach called MSRIHL [1] integrates low-level Haar-like features into deep learning models for accurate SAR object detection, highlighting its potential effectiveness.

**Wavelet Scattering Transform [45] (WST).** The WST is a powerful signal-processing technique that is widely used in image processing. It aims to extract robust and discriminative low-level and high-level features simultaneously. By capturing both high-frequency and low-frequency information, it provides a rich representation of both local and global image features. The hierarchical representation offered by the Wavelet Scattering Transform enables feature extraction at different scales and resolutions, which is particularly useful for capturing fine details of small objects in SAR images and robustly handling high-frequency noise. Vidal et al. [61] conduct a comprehensive investigation demonstrating the high potential performance of Wavelet Transform in SAR image denoising. Many recent works [93; 26; 50; 78] utilize Wavelet Transform or WST for robust feature extraction and integration with CNN networks for target recognition. These studies validate the feasibility of incorporating WST features into modern CNN models.

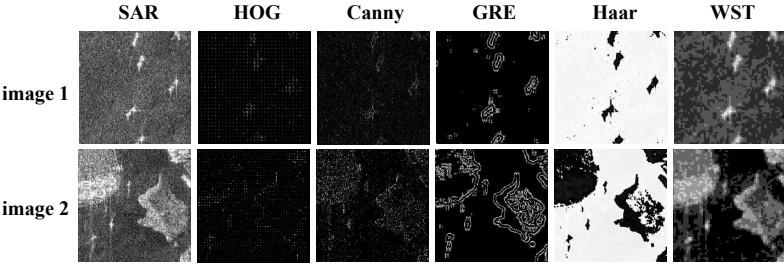

Figure S7: Visualization of handcrafted features on SAR images. (To facilitate visualization, the features are average pooled and represented as a single channel.)

## A.4 Multi-Stage Filter-Augmentation

Fig. S8 provides a visual illustration of the proposed filter-augmented data input. It involves concatenating the original single-channel grayscale Synthetic Aperture Radar (SAR) image, denoted as $x$, with the filter-augmentation representation $M_i^x$.

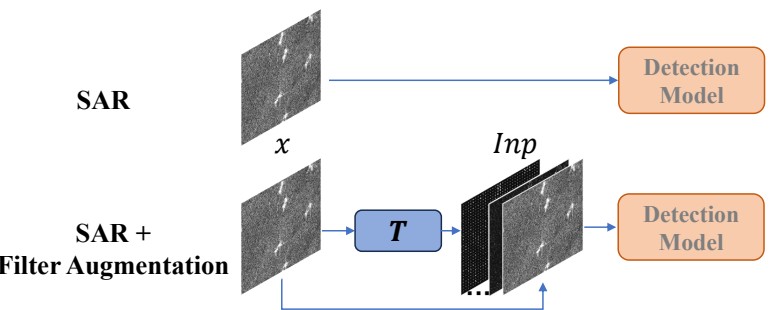

Figure S8: Illustration of filter-augmented input.

Table S7: Comparsion of different filter augmented inputs using Faster R-CNN and Resnet-50 as detection model.

| Modality | mAP | @50 | @75 | @s | @m | @l |
|---|---|---|---|---|---|---|
| SAR (as RGB) | 50.2 | 83.0 | 54.8 | 44.8 | 61.6 | 58.6 |
| SAR+Canny | 50.7 | 83.6 | 55.0 | 45.3 | 62.0 | 57.1 |
| SAR+Hog | 50.7 | 83.5 | 55.2 | 45.1 | 61.4 | **58.4** |
| SAR+Haar | 50.6 | 83.4 | 54.7 | 45.4 | 61.6 | 58.0 |
| SAR+WST | **51.1** | 83.9 | 54.7 | 45.2 | **62.3** | 57.5 |
| SAR+GRE | 50.6 | 83.8 | 54.7 | 44.8 | 61.7 | 57.6 |
| SAR+Hog+Haar+WST | **51.1** | **84.0** | **55.9** | **45.7** | 62.0 | 58.2 |

Table S8: Generalization of MSFA on different detection frameworks. INP: Traditional ImageNet Pretrain on backbone network only.

| Framework | | Pretrain | Test | | | | | |
|---|---|---|---|---|---|---|---|---|
| | | | **mAP** | @50 | @75 | @s | @m | @l |
| Two Stage | Faster RCNN [53] | INP | 49.0 | 82.2 | 52.9 | 43.5 | 60.6 | 55.0 |
| | | MSFA | 51.1 (+2.1) | 83.9 | 54.7 | 45.2 | 62.3 | 57.5 |
| | Cascade RCNN [4] | INP | 51.1 | 81.9 | 55.8 | 44.9 | 62.9 | 60.3 |
| | | MSFA | 53.9 (+2.8) | 83.4 | 59.8 | 47.2 | 66.1 | 63.2 |
| | Grid RCNN [4] | INP | 48.8 | 79.1 | 52.9 | 42.4 | 61.9 | 55.5 |
| | | MSFA | **51.5** (+2.7) | 81.7 | 56.3 | 45.1 | 64.1 | 60.0 |
| Single Stage | RetinaNet [33] | INP | 47.4 | 79.3 | 49.7 | 40.0 | 59.2 | 57.5 |
| | | MSFA | 49.0 (+1.6) | 80.1 | 52.6 | 41.3 | 61.1 | 59.4 |
| | GFL [30] | INP | 49.8 | 80.9 | 53.3 | 42.3 | 62.4 | 58.1 |
| | | MSFA | 53.7 (+3.9) | 84.2 | 57.8 | 47.8 | 66.2 | 59.5 |
| | FCOS [59] | INP | 46.5 | 80.9 | 49.0 | 41.1 | 59.2 | 50.4 |
| | | MSFA | 48.5 (+2.0) | 82.1 | 51.4 | 42.9 | 60.4 | 56.0 |
| End to End | DETR [6] | INP | 31.8 | 62.3 | 30.0 | 22.2 | 44.9 | 41.1 |
| | | MSFA | 47.2 (+15.4) | 77.5 | 49.8 | 37.9 | 62.9 | 58.2 |
| | Deformable DETR [94] | INP | 50.0 | 85.1 | 51.7 | 44.0 | 65.1 | 61.2 |
| | | MSFA | 51.3 (+1.3) | 85.3 | 54.0 | 44.9 | 65.6 | 61.7 |
| | Sparse RCNN [58] | INP | 38.1 | 68.8 | 38.8 | 29.0 | 51.3 | 48.7 |
| | | MSFA | 41.4 (+3.3) | 74.1 | 41.8 | 33.6 | 53.9 | 53.4 |
| | Dab-DETR [39] | INP | 45.9 | 79.0 | 47.9 | 38.0 | 61.1 | 55.0 |
| | | MSFA | 48.2 (+2.3) | 81.1 | 51.0 | 41.2 | 63.1 | 55.4 |

Table S9: Generalization of MSFA on different detection backbones. INP: Traditional ImageNet Pretrain on backbone network only.

| Framework | #P(M) | Pretrain | Test | | | | | |
| --- | --- | --- | --- | --- | --- | --- | --- | --- |
| | | | **mAP** | @50 | @75 | @s | @m | @l |
| R50 [23] | 25.6 | INP | 49.0 | 82.2 | 52.9 | 43.5 | 60.6 | 55.0 |
| | | MSFA | 51.1 (+2.1) | 83.9 | 54.7 | 45.2 | 62.3 | 57.5 |
| R101 [23] | 44.7 | INP | 51.2 | 84.1 | 55.6 | 45.9 | 61.9 | 56.3 |
| | | MSFA | 52.0 (+0.8) | 84.6 | 56.6 | 46.6 | 63.4 | 57.7 |
| R152 [23] | 60.2 | INP | 51.9 | 85.2 | 55.9 | 46.4 | 62.5 | 57.9 |
| | | MSFA | 52.4 (+0.5) | 85.4 | 57.2 | 47.4 | 63.3 | 58.7 |
| ConvNext-T [42] | 28.6 | INP | 53.2 | 86.3 | 58.1 | 47.2 | 65.2 | 59.6 |
| | | MSFA | 54.8 (+1.6) | 87.1 | 59.8 | 48.8 | 66.7 | 62.1 |
| ConvNext-S [42] | 50.1 | INP | 54.2 | 87.8 | 59.2 | 49.2 | 65.8 | 59.8 |
| | | MSFA | 55.4 (+1.2) | 87.6 | 60.7 | 50.1 | 67.1 | 61.3 |
| ConvNext-B [42] | 88.6 | INP | 55.1 | 87.8 | 59.5 | 48.9 | 66.9 | 61.1 |
| | | MSFA | 56.4 (+1.3) | 88.2 | 61.5 | 51.1 | 68.3 | 62.4 |
| VAN-T [22] | 4.1 | INP | 45.8 | 79.8 | 48.0 | 38.6 | 57.9 | 53.3 |
| | | MSFA | 47.6 (+1.8) | 81.4 | 50.6 | 40.5 | 59.4 | 56.7 |
| VAN-S [22] | 13.9 | INP | 49.5 | 83.8 | 52.8 | 43.2 | 61.6 | 56.4 |
| | | MSFA | 51.5 (+2.0) | 85.0 | 55.6 | 44.8 | 63.4 | 60.4 |
| VAN-B [22] | 26.6 | INP | 53.5 | 86.8 | 58.0 | 47.3 | 65.5 | 60.6 |
| | | MSFA | 55.1 (+1.6) | 87.7 | 60.2 | 48.8 | 67.3 | 62.2 |
| Swin-T [41] | 28.3 | INP | 48.4 | 83.5 | 50.8 | 42.8 | 59.7 | 55.7 |
| | | MSFA | 50.2 (+1.8) | 84.1 | 53.9 | 44.1 | 61.3 | 58.8 |
| Swin-S [41] | 49.6 | INP | 53.1 | 87.3 | 57.8 | 47.4 | 63.9 | 60.6 |
| | | MSFA | 54.0 (+0.9) | 87.0 | 59.2 | 48.2 | 64.5 | 61.9 |
| Swin-B [41] | 87.8 | INP | 53.8 | 87.8 | 59.0 | 49.1 | 64.6 | 60.0 |
| | | MSFA | 55.7 (+1.9) | 87.8 | 61.4 | 50.5 | 66.5 | 62.5 |

## A.5 Experimental Results

### A.5.1 Main results on SARDet-100K.

The detailed experimental outcomes are presented in Table S7, Table S8, and Table S9, where comprehensive, fine-grained testing metrics are provided. These include AP@50, AP@75, AP@small (AP@s), AP@medium (AP@m), and AP@large (AP@l), offering deeper insights and robust results for model evaluation.

Table S7 serves as an extension of Table 3 from the main paper, showcasing a comparison of various filter-augmented inputs utilizing Faster R-CNN with ResNet-50 as the detection model.

Table S8 expands upon Figure 6(a) of the main paper, and Table S9 expands upon Figure 6(b). These tables centre on the experiments that explore the generalizability of the MSFA framework across different detection frameworks and backbones. Notably, a significant enhancement in performance is observed across various frameworks, including single-stage, two-stage, and query-based frameworks, as well as across a spectrum of backbone architectures, ranging from ResNets and modern-designed ConvNets to Vision Attention Networks (VANs) and Vision Transformer (ViT)-based Swin networks. These results offer compelling evidence of the efficacy and broad applicability of the method we propose.

Table S10: Dataset statistics comparison between DOTA and DIOR. *: multi-scale preprocessed.

| Dataset | Images | Instances | Categories | Image Size | Avg. Instance Area |
|---------|--------|-----------|------------|------------|--------------------|
| DOTA* | 68,324 | 1,058,641 | 15 | 1024*1024 | 5,021 |
| DIOR | 23,463 | 192,518 | 20 | 800*800 | 12,726 |

Table S11: Comparison of MSFA and Finetuning Performance under similar computational budgets. INP: Backbones are pretrained on ImageNet.

| Model | INP | MSFA Epoch | Finetune Epochs | Total Epochs | Total Iterations | mAP |
|-------|-----|------------|-----------------|--------------|------------------|-----|
| Faster-RCNN [53] | ✓ | 12 | 24 | 36 | 16.1k | 54.5 |
| Faster-RCNN [53] | ✓ | 0 | 36 | 36 | 17.7k | 52.8 |

To ensure a fair comparison, we evaluate MSFA and vanilla detection models under similar computational budgets. In Table S11, we train Faster-RCNN models (with a ResNet-50 ImageNet-1K pretrained backbone) with 12 epochs of MSFA pretraining on the DOTA dataset, followed by finetuning on the SARDet-100K dataset. This is compared to directly finetuning the model for 36 epochs on the SARDet-100K dataset. With the same total number of training epochs and a slightly less total number of iterations, the proposed MSFA achieves a significantly higher (1.7% higher) mAP result on the SARDet-100K test set. MSFA pretraining incorporates general knowledge from both natural and remote sensing datasets, which allows efficient and effective finetuning and helps mitigate overfitting in downstream tasks. It is also important to note that, MSFA pretraining is a one-time effort. The pretrained MSFA models can be reused for finetuning different SAR detection datasets.

### A.5.2   Comparison of SARDet-100K with other datasets.

To assess the quality of the proposed SARDet-100K dataset as a large-scale SAR object detection benchmark, we evaluate different models on this dataset and compare the results with those obtained from other popular benchmark datasets, such as SSDD [84] and HRSID [71]. The results are presented in Fig S9. These results indicate that the performance of modern models on our dataset has not yet reached saturation. Among the various models reviewed, there is an 8.4% performance gap between the weakest and strongest models, whereas, for SSDD and HRSID, this gap is only 4.1% and 4.3%, respectively. This suggests that SSDD and HRSID are relatively simple for most existing models, leading to performance saturation on these smaller datasets.

Furthermore, we observed that on these smaller datasets, models with a relatively large number of parameters tend to suffer from performance degradation due to overfitting. For example, ResNet-152 underperforms compared to ResNet-101 on SSDD, and ResNet-101 underperforms compared to ResNet-50 and ResNet-18 on HRSID. However, on the large-scale SARDet-100K, this issue does not arise. On our larger dataset, models continue to benefit from increased model size, indicating that our proposed dataset is suitable for developing relatively larger models for large-scale SAR object detection.

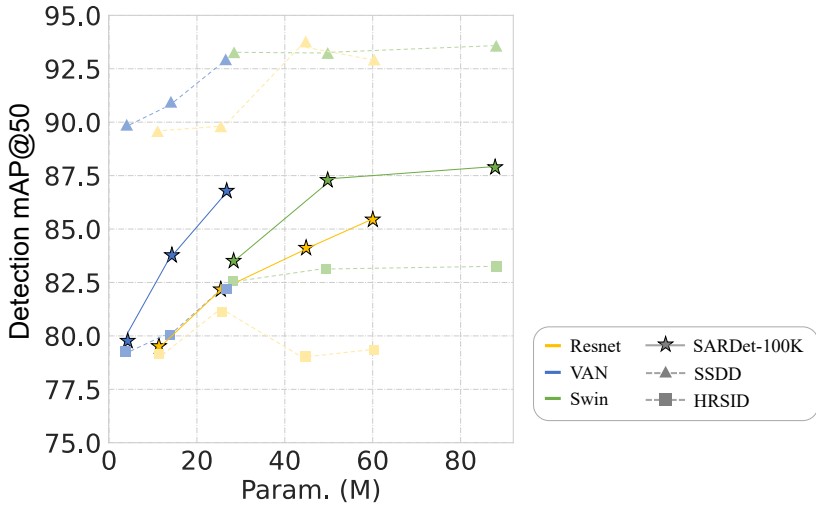

Figure S9: Evaluate different backbone models on SARDet-100K dataset and other previous popular benchmarks (SSDD [84] and HRSID [71]). Backbones are plugged under Faster-RCNN [53] detection framework.

## A.6 Implementaion Details

For the ImageNet pretrain, we adopt a 100-epoch backbone pretraining strategy on the Imagenet-1K, with their default training strategy in MMPretrain [11] configurations.

For the second stage of the proposed multistage pretrain, we select the large-scale optical remote sensing DOTA dataset as the main dataset. We also do comparison experiments on the DIOR dataset to find out the downstream performance affection factors on the second stage pretrain dataset. Their detailed information is illustrated in Table S10.

For the DOTA dataset, we perform an extra dataset preprocess because DOTA dataset images have a large range of different image resolutions. The original image contains only 1,411 training images. To have more image and multiscale instances for effective training, we follow [31] to adopt a multi-scale dataset splitting strategy by rescaling the original high-resolution images into three distinct scales (x0.5, x1.0, x1.5), and then cropping each scaled image into 1024×1024 patches with an overlap of each patch of 500 pixels, to avoid destruction division on the instance at patch borders. With the preprocessed dataset, load resize as 1024*1024, with a RandomFlip probability of 0.5. For the DIOR dataset, we train the model by resizing the images as 800*800, with a RandomFlip probability of 0.5.

For finetuning on SARDet-100k, SSDD and HRSID, we train by resizing the image into 800*800, with a RandomFlip probability of 0.5. We train the model by 12 epochs on the train set and test the model on the test set with the 12th epoch checkpoint.

We primarily conduct our experiments using the MMPretrain [11] and the MMDetection [8] frameworks, on 8 RTX-3090 GPUs (24G). For detailed information on the hyperparameters and training settings, please refer to Table S12.

## A.7 Detection Result Visualizations

The detection visualization results, comparing the MSFA framework with the traditional ImageNet pretraining approach, are presented in Fig. S10. These results demonstrate that MSFA outperforms the conventional ImageNet backbone pretrain in terms of reducing missed detections, false detection, and improving localization accuracy.

Table S12: Hyper-parameter of pretrain and finetune settings. Cls.: Classification, Det.: Detection, B.S.: Batch Size, L.R.: Learning Rate.

| Task / Model | Dataset | Optim. | B.S. | L.R | Epochs |
|---|---|---|---|---|---|
| Cls. Pretrain | ImageNet | AdamW | 512 | 1e-8 | 100 |
| Det. Pretrain | DOTA | AdamW | 16 | 1e-4 | 12 |
| Det. Pretrain | DIOR | AdamW | 16 | 1e-4 | 12 |
| Det. Finetune | SARDet-100k | AdamW | 16 | 1e-4 | 12 |
| Det. Finetune | SSDD | AdamW | 32 | 2.5e-4 | 12 |
| Det. Finetune | HRSID | AdamW | 32 | 2.5e-4 | 12 |
| DETR | DOTA/SARDet-100k | AdamW | 16 | 1e-4 | 150 |
| Deformable-DETR | DOTA/SARDet-100k | AdamW | 16 | 2e-4 | 50 |
| Dab-DETR | DOTA/SARDet-100k | AdamW | 16 | 1e-4 | 50 |
| Sparse-RCNN | DOTA/SARDet-100k | AdamW | 16 | 2.5e-5 | 12 |

Table S13: ConvNext-B MSFA

| Category | mAP | @50 | @75 | @s | @m | @l |
|---|---|---|---|---|---|---|
| ship | 0.669 | 0.923 | 0.783 | 0.653 | 0.714 | 0.555 |
| aircraft | 0.455 | 0.753 | 0.466 | 0.419 | 0.458 | 0.47 |
| car | 0.655 | 0.985 | 0.792 | 0.553 | 0.674 | n/a |
| tank | 0.454 | 0.766 | 0.427 | 0.429 | 0.891 | n/a |
| bridge | 0.441 | 0.885 | 0.368 | 0.411 | 0.609 | 0.714 |
| harbor | 0.711 | 0.979 | 0.858 | 0.601 | 0.751 | 0.756 |
| Average | 0.564 | 0.882 | 0.616 | 0.511 | 0.733 | 0.624 |

## A.8 Failure Scenarios

However, the current model is not without its drawbacks and imperfections. Fig. S11 highlights several failure scenarios. When the input SAR images lack identifiable details or contextual information, it can lead to incorrect classifications. When SAR images contain small and densely packed objects, the model may fail to detect some of them. Poor image quality characterized by fading, blurriness, or overall low resolution, further exacerbates the risk of missed detections. In challenging cases, the detector may also struggle with accurate localization.

Regarding fine-grained category detection performance, Table S13 presents the detection results of Faster-RCNN [53] with ConvNext-B [42] and MSFA pretraining. Notably, the model exhibits relatively low performance for objects with:

- Small sizes, such as Tank (with an average area of 461 pixels)
- Large length-width ratios, such as Bridge
- High appearance variability, such as Airplane

However, the primary focus of this work is to address the domain and model gaps existing in SAR object detection pretraining and finetuning. Our method demonstrates excellent compatibility with most existing deep networks and can seamlessly integrate with models specifically designed to tackle the above challenging scenarios.

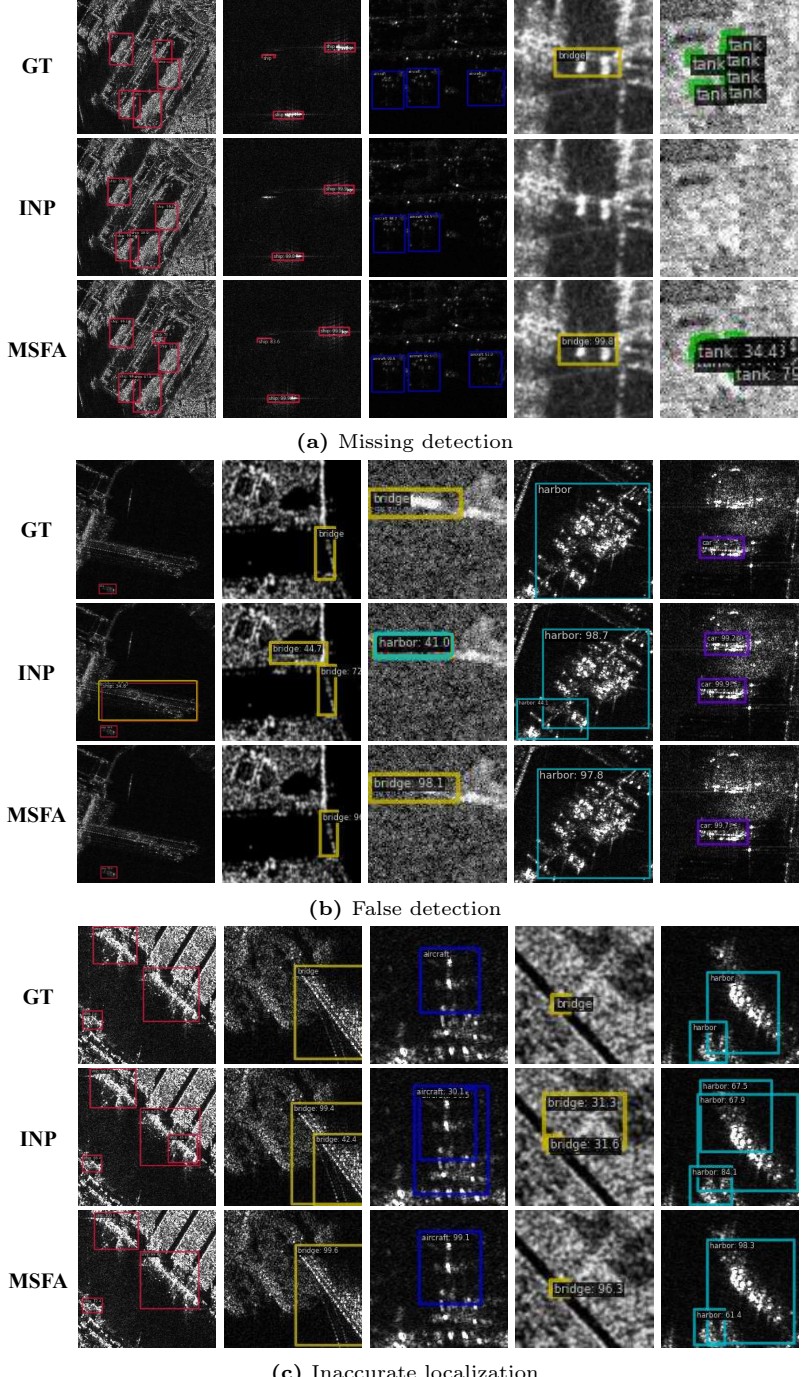

Figure S10: MSFA better than traditional ImageNet backbone pretrain in (a) missing detection, (b) false detection and (c) inaccurate localization

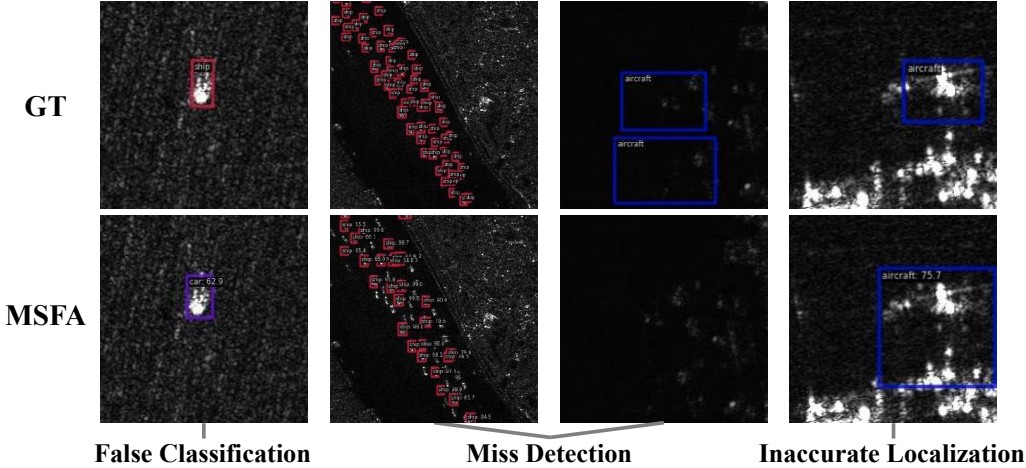

Figure S11: Visulization of failure cases: false classification, miss detection, inaccurate localization.

