# OpenReview forum: "SARDet-100K: Towards Open-Source Benchmark and ToolKit for Large-Scale SAR Object Detection"
_NeurIPS.cc/2024/Conference — NeurIPS 2024 spotlight_

### Official Review · Reviewer_1iNV · 2024-06-19

**Soundness:** 4
**Presentation:** 4
**Contribution:** 4
**Rating:** 8
**Confidence:** 5

**Summary:**

The authors address the limitations of existing datasets and the inaccessibility of source codes by creating a new benchmark dataset, SARDet-100K, which is a large-scale, multi-class dataset. Additionally, the paper proposes a Multi-Stage with Filter Augmentation pretraining framework designed to overcome the domain and model gaps between pretraining on RGB datasets and finetuning on SAR datasets. The MSFA method demonstrates effectiveness and generalizability across various models.

**Strengths:**

1. The creation of the SARDet-100K dataset is a significant contribution. It offers the research community a large-scale, diverse dataset that was previously lacking.
2. As I know, in the field of SAR object detection, open-source code is indeed rare, which has significantly hindered the progress of development. It is very exciting to see a well-documented and professional open-source code base. Making the dataset and code publicly available enhances the reproducibility of the research and facilitates further innovation by other researchers.
3. Most previous work on improving SAR object detection performance focus on designing neural network modules. It is interesting that this research tackles the problem from the perspective of pretraining and domain transition.
4. The paper provides sufficient and detailed experiments/analysis that validate the effectiveness of the proposed MSFA method.

**Weaknesses:**

1. You mentioned that optical remote sensing datasets (like DOTA) shares similar object shapes, scales, and categories in SAR datasets, therefore the downstream SAR datasets detection can benefits from the transferred knowledge. But why not discuss joint train the DOTA and SARDet-100K datasets together? In this way, the model can also learn joint representation from both DOTA and SARDet-100K therefore potentially improves the detection performance on SARDet-100K.
2. It is recommended to add a few more recent methods for comparison in Table 5.

**Questions:**

See above.

**Limitations:**

Limitations are properly addressed.

---

> ### Author Rebuttal · Authors · 2024-08-01
>
> We appreciate the reviewer's insightful comments and suggestions. We have carefully considered the points raised and provide the following clarifications and additions:
>
> While it is true that merging multiple datasets can improve performance in **general object detection** tasks where datasets share similar concepts (e.g., optical concepts), this approach has been shown to improve performance due to shared features and representations. However, in our scenario, SAR and optical (RGB) datasets represent different modalities, which present distinct challenges:
>
> SAR and RGB datasets differ significantly in terms of data modality and conceptual representation. Joint training of these datasets can result in a crowded feature and representation space, limiting the model's expressiveness for SAR data.
>
> The distinct nature of SAR and RGB data leads to different learning difficulties, causing unsynchronized optimization rates across various parts of the model. Inconsistent optimization objectives across modalities can cause conflicting optimization directions, further reducing the performance of SAR object detection.
>
> To justify our analysis, we conducted experiments using ConvNext-T backbones pretrained on ImageNet. The results are presented below:
> | Train on                            | Test on    | mAP | 50  | 75  |
> |-----------------------------------|------------  |-----    |----- |----- |
> | SARDet-100K                       | SARDet-100K| 53.2| 86.3| 58.1|
> | DOTA                              | DOTA       | 45.2| 70.4| 49.1|
> | SARDet-100K + DOTA                | SARDet-100K| 51.9| 83.2| 56.7|
> | SARDet-100K + DOTA                | DOTA       | 42.4| 67.1| 46.1|
> | (DOTA pretrain) SARDet-100K     | DOTA       | 54.8| 87.1| 59.8|
>
> The table demonstrates that joint training on SARDet-100K and DOTA results in a decreased mAP for SARDet-100K (from 53.2 to 51.9). In contrast, our proposed pretraining on DOTA before training on SARDet-100K improves the performance (mAP of 54.8). Additionally, the DOTA dataset's performance also drops significantly with joint training, supporting our claim.
>
> We have incorporated additional recent works for comparison in Table 5 to provide a more comprehensive evaluation. The updated table is shown in Table R1 (in the rebuttal pdf).

---

> > ### Comment · Reviewer_1iNV · 2024-08-11
> >
> > The authors have solved my concerns, I am convinced to maintain my initial judgment.

---

### Official Review · Reviewer_qv3Q · 2024-07-04

**Soundness:** 3
**Presentation:** 3
**Contribution:** 3
**Rating:** 7
**Confidence:** 5

**Summary:**

The authors establish a new benchmark SAR object detection dataset (SARDet-100K) and open-source SAR detection pretrain method (MSFA). This initiative significantly addresses the limitations posed by the scarcity of public SAR datasets and the inaccessibility of source codes, fostering further research and development in SAR object detection.

**Strengths:**

Providing a larger standardized dataset for the data-scarce field of SAR target detection addresses a critical need and significantly contributes to its development. The authors unify and standardizes ten existing datasets to create SARDet-100K, the first COCO-level large-scale dataset for SAR multi-category object detection, which represents a substantial effort.

MSFA model proposed in this paper is both effective and concise. It is refreshing that the MSFA model ingeniously applies traditional handcrafted features instead of design-heavy deep learning methods. Moreover, unlike previous approaches that use handcrafted features for feature refinement in deep learning, this work employs them for model pre-training and domain transformation, representing a novel and innovative approach.

**Weaknesses:**

The paper does not clarify if the metrics reported on the SARDet-100K dataset are for the test set or the validation set. Additionally, please clarify the training setting: is the model checkpoint used for testing from the best validation or from the last epoch?

As a benchmark dataset and method, actual runtimes and memory usage should be reported. These details are currently missing from the paper.

There is a lack of clarification on the abbreviations used in Table 1 and S12.

**Questions:**

Regarding the image slicing in your dataset standardization process, how do you handle objects that lie on the slicing border? Will the slicing split the objects apart?

**Limitations:**

Limitations are well discussed in Section 6 and A.8.

---

> ### Author Rebuttal · Authors · 2024-07-31
>
> Sorry for the lack of clarity in the manuscript. The reported performance metrics are based on the test set of the SARDet-100K dataset. The models are trained on the training set for a total of 12 epochs, and the checkpoint from the 12th epoch are used for testing.
>
> We acknowledge the importance of reporting actual runtimes and memory usage. In response, we will release all training checkpoints and logs as part of our open-source code repository. These logs will include detailed information such as actual runtimes, memory usage, hardware specifications, system environment, package versions, and training loss, ensuring full transparency and reproducibility.
>
> Sorry for any confusion caused by the abbreviations used in Table 1 and S12. Here are the clarifications:
> Ins.: Instances Img.: Images Cls.: Classification Det.: Detection B.S.: Batch Size L.R.: Learning Rate.
>
> Thank you for pointing out this practical issue. In our dataset standardization process, we handle objects that lie on the slicing border as follows: If at least 50% of the object’s area remains within the image slice, we keep the bounding box; otherwise, we ignore the bounding box and treat it as background. For the kept bounding boxes, we retain the original full box coordinates, meaning that the annotation coordinates for such objects may extend beyond the image boundaries.
>
> We will add all the above details and clarifications to the revised version of the paper.

---

### Official Review · Reviewer_paPp · 2024-07-05

**Soundness:** 3
**Presentation:** 3
**Contribution:** 3
**Rating:** 7
**Confidence:** 5

**Summary:**

This study presents the a large-scale dataset designed for SAR object detection, alongside a Multi-Stage with Filter Augmentation pretraining framework. The authors address the challenges associated with the limited availability of public SAR datasets and the lack of accessible source codes. The proposed method is effective and can be generalized to most modern backbone and detection networks.

**Strengths:**

The establishment of the SARDet-100K dataset provides a robust foundation for large-scale, multi-class SAR object detection research. Moreover, the open sourcing of the SAR detection codebase significantly enhances research reproducibility. The introduction of the MSFA pretraining framework is a novel approach that effectively bridges the domain and model gaps between RGB and SAR datasets by leveraging traditional handcrafted features. Overall, the paper is well-written, and the experimental validation and analysis are sound and solid.

**Weaknesses:**

Your novelty lies in incorporating handcrafted features, the introduction and related work on these features are somewhat lacking. Given the current dominance of deep learning methods, many junior researchers may not be familiar with classic handcrafted feature descriptors. Therefore, it would be beneficial to provide a more comprehensive introduction and conceptual visualization for each of the mentioned handcrafted features to enhance the paper's clarity and accessibility.

The supplementary materials would benefit from including a clear, step-by-step guide on training and testing the models. While the code is provided, the lack of direct scripts and instructions makes it difficult to replicate the results.

The category distribution in Figure S6(b) reveals SARDet-100K is a significantly imbalanced dataset. This imbalance may lead to the long-tail problem, potentially hindering the performance of models on tail categories. Why do not you consider balancing the dataset?

**Questions:**

In Figure 1, are the compared image pairs spatially aligned? It seems not. Would it be an issue?

**Limitations:**

None.

---

> ### Author Rebuttal · Authors · 2024-07-31
>
> We appreciate your feedback regarding the introduction and related work on handcrafted feature descriptors. We have indeed provided more extensive related work on these descriptors in the Appendix section of our submission. In the revised version of the paper, we will include comprehensive visualizations of different handcrafted feature descriptors (including HOG, Canny, GRE, Haar and WST) to enhance clarity and accessibility, as Figure R1 (in the rebuttal pdf).
>
> Thank you for pointing out the need for a clearer guide in the supplementary materials. Our code is based on the open-source mmdetection framework, and a detailed guide can be found in the official mmdetection GitHub repository. However, we acknowledge that a more specific guide related to our work would be beneficial. Therefore, we will include a clear, step-by-step guide on training and testing the models directly in our open-sourced code to facilitate easier replication of our results.
>
> The category distribution in Figure S6(b) indeed reveals an imbalance in the SARDet-100K dataset. We intentionally maintained this long-tail distribution to simulate real-world applications of large-scale multi-category SAR object detection, where such imbalances commonly exist. Addressing the long-tail problem is a significant challenge and an active area of research. Our work aims to provide a realistic benchmark for future studies that may focus on developing techniques to address these imbalances effectively.
>
> Yes, the image pairs in Figure 1 are not spatially aligned, except for the second column. This misalignment is due to the difficulty in finding publicly available SAR-RGB paired images. The purpose of Figure 1 is to provide a conceptual overview, and the lack of spatial alignment does not hinder the understanding of the key concepts being presented.

---

> > ### Comment · Reviewer_paPp · 2024-08-09
> >
> > The authors have addressed all the concerns, and I would like to raise my score to accept.

---

### Official Review · Reviewer_qrry · 2024-07-10

**Soundness:** 3
**Presentation:** 3
**Contribution:** 4
**Rating:** 7
**Confidence:** 5

**Summary:**

This work introduces SARDet-100K, a new large-scale, multi-category dataset for SAR object detection. It also proposes a novel Multi-Stage with Filter Augmentation pretraining framework to mitigate domain and model gaps encountered when transferring models pretrained on RGB datasets to SAR datasets. A new benchmark dataset and open-source method in SAR object detection is established.

**Strengths:**

The paper's most significant contribution is the creation of the SARDet-100K dataset, which is the first large-scale, multi-category benchmark for SAR object detection. This dataset addresses the long-standing issue of limited and homogeneous SAR datasets, providing a rich resource that is likely to stimulate further research and development. The introduction of the MSFA pretraining framework is another strength, as it effectively addresses the domain and model gaps between RGB and SAR imagery, demonstrating robust performance across various deep learning models.
Additionally, the paper is well-written, clearly outlining the motivation, methodology, and implications of the research, making it accessible to a broad audience in the field.

**Weaknesses:**

The concept of the 'model gap' is not clearly defined in the current manuscript. Could you provide a more detailed explanation of what you mean by 'model gap' within the context of your study?

Instead of a time-consuming pretraining stage, why not directly train the SAR detection model using only the proposed dataset with sufficiently long iterations since the dataset already contains a sufficient amount of images and instances.

**Questions:**

None

**Limitations:**

Limitations are well discussed.

---

> ### Author Rebuttal · Authors · 2024-08-01
>
> In the context of our study, the term 'model gap' refers to the inconsistency introduced when only the backbone of the detector is pre-trained, while other components, such as the neck and heads, are initialized randomly. For downstream object detection tasks, the model comprises the entire detector, which includes the backbone, neck, and heads. When performing object detection, initializing only the backbone while randomly initializing the other components creates a mismatch or gap between the pre-trained weights and the fine-tuned detector network. This inconsistency, referred to as the 'model gap,' can negatively impact the overall performance of the detector as the synergy between the backbone and other components is disrupted.
>
> We have considered the approach of directly training the SAR detection model on the proposed dataset. However, there are two main reasons for opting for the pretraining method:
>
> 1. Versatility of Pretrained Models: Our proposed method involves pretraining the model once, which can then be utilized for object detection across multiple SAR datasets. For instance, in our study, we pre-trained the model using the MSFA framework and demonstrated its applicability on the SARDet-100K, SSDD, and HRSID datasets. This 'one-for-all' pretraining approach saves considerable time during the fine-tuning stage, as the pretrained model serves as a robust starting point for various datasets.
>
> 2. Mitigation of Overfitting: Pretraining helps reduce the risk of overfitting. As illustrated in Table S11 of our manuscript, pretraining on the DOTA dataset for 12 epochs followed by fine-tuning for 24 epochs (totaling 16.1K iterations) resulted in an mAP of 54.5. In contrast, directly fine-tuning for 17.7K iterations without pretraining yielded a lower mAP of 52.8. This demonstrates that the pretraining stage not only enhances performance but also provides a more efficient training process.

---

> > ### Comment · Reviewer_qrry · 2024-08-12
> >
> > Thanks for your reply. The authors have addressed the issues. I tend to accept it.

---

### Author Rebuttal · Authors · 2024-08-01

We add Figure R1, comprehensive visualizations of different handcrafted feature descriptors (including HOG, Canny, GRE, Haar and WST) to enhance clarity and accessibility.

We revise the main paper Table 5 to Table R1 (Table for comparison of the proposed MSFA with previous state-of-the-art methods on SSDD and HRSID datasets):  We incorporate additional recent works for comparison to provide a more comprehensive evaluation.

---

### Decision · Program_Chairs · 2024-09-25

**Decision:**

Accept (spotlight)

**Comment:**

All reviewers recognize the contribution of this submission. After rebuttal, the authors provide further clarification of the issues mentioned by the reviewers. Now, all reviewers have no further issue for the submission. The AC agrees the reviewers' comments and learns toward an acceptance for this paper.